# The Attitude of Portuguese Physical Education Teachers toward Physical Fitness

**DOI:** 10.3390/children9071005

**Published:** 2022-07-04

**Authors:** Adilson Marques, Diogo Balsa, Marta Domingos, Rafael Cavalheiro, Tiago Carreira, Tiago Moreira, Tiago Ribeiro, Élvio R. Gouveia

**Affiliations:** 1CIPER, Faculdade de Motricidade Humana, Universidade de Lisboa, 1499-002 Lisbon, Portugal; amarques@fmh.ulisboa.pt; 2Instituto de Saúde Ambiental (ISAMB), Faculdade de Medicina, Universidade de Lisboa, 1649-028 Lisboa, Portugal; 3Faculdade de Motricidade Humana, Universidade de Lisboa, 1499-002 Lisbon, Portugal; diogo_balsa95@outlook.com (D.B.); martapdomingos@gmail.com (M.D.); ravlcavalheiro@gmail.com (R.C.); tiago.carreira.771@gmail.com (T.C.); tiagoliveiramoreira@gmail.com (T.M.); tiagodelgadoribeiro@gmail.com (T.R.); 4Department of Physical Education and Sport, University of Madeira, 9000-390 Funchal, Portugal; 5LARSYS, Interactive Technologies Institute, 9020-105 Funchal, Portugal

**Keywords:** physical fitness, fitness tests, physical education, attitudes of teachers

## Abstract

In this study, we aimed to examine how Portuguese physical education teachers perceive the development of physical fitness through fitness tests in schools. The participants were 764 Portuguese teachers teaching at middle-school and high-school levels. The Physical Education Teacher Attitudes Toward Fitness Tests Scale (PETAFTS) was used to collect the data. The means and confidence intervals for each attitude subdomain and the overall attitude were computed. A one-way ANOVA was used to examine the group differences in three subdomains of the attitudes of teachers by different variables. The overall attitude of teachers toward fitness tests was slightly positive according to a 7-point Likert scale (5.52, 95% CI: 5.47, 5.58). The results suggested that female teachers found fitness tests more useful, but male teachers significantly enjoyed implementing them. The data collected also showed that younger teachers found the implementation of fitness tests significantly more enjoyable than older teachers. In conclusion, future research should prioritise specific intervention content considering gender and the age of teachers to reinforce the development of physical fitness through fitness tests in schools.

## 1. Introduction

Promoting physical activity (PA) and healthy lifestyles has become a worldwide priority for public health authorities [1]. Schools are a critical context for acting with children and adolescents to achieve this priority. Through physical education (PE), schools provide an opportunity for youths to be physically active and to promote healthy lifestyles.

According to the physical activity recommendations, children and adolescents from 5 to 17 years old should accumulate at least 60 min of moderate-to-vigorous-intensity physical activity daily, mostly aerobic [2]. Moreover, PA should promote physical fitness (PF) and wellbeing, enhancing a healthy lifestyle that can increase self-esteem and prevent injuries [3]. PF is a concept that has gained interest in recent research. The definition and construct of PF can be defined as the ability to perform daily tasks without undue fatigue, having sufficient energy to enjoy leisure-time activities, and the ability to meet unforeseen emergencies [4]. There is evidence linking low levels of PF to an increase in cardiovascular disease risk factors in children and adolescents [5]. PF is a health marker in childhood and adolescence [6,7].

Recent research suggests that the PF levels of students in PE have decreased over the years [8]. PF levels tend to transit from childhood to adolescence [9] and to adulthood [10], which supports and reinforces the importance of developing optimal levels from an early age [11]. Thus, it is important to introduce PF components into PE lessons as part of the curriculum [12]. Fitness testing has been integrated in PE programmes for more than half a century [13]. Nowadays, numerous countries have fitness testing integrated into their PE curriculum and others (e.g., Russia and China) are also looking to implement them [14,15]. However, there are a few controversial ideas whether PF tests should be part of the PE curriculum. Existing organisations and authors have found that school-based fitness tests are essential to gather information; others focus on the lack of validity of these tests [16]. Fitnessgram is the most widely used fitness software system to assess PF in schools [17] and has proliferated in American schools since the 1950s [18]. According to the literature, Eurofit is probably the most popular way to measure physical fitness in Europe [19], and is known as a reliable test battery to assess PF levels in various populations [20]. The Portuguese PE curriculum has a significant emphasis on PF tests, which are used in every school of the country. The instrument used in Portugal, FITescola, is a digital platform that monitors the fitness levels of students mostly through field tests and tries to promote health in youths [21].

As highlighted above, the importance of promoting PA and developing PF at an early age, especially when related to fitness tests in PE lessons, leads to controversial perspectives. The attitude of PE teachers toward fitness tests and whether they should or should not be included in the PE curriculum is not consensual [22]. This may be due to the lack of knowledge teachers have regarding health-related fitness, suggesting a necessary investment in training and development in this area [23]. Promoting a healthy lifestyle and creating habits so that children cultivate a positive attitude toward PA and PF from an early age can also be debatable because there is little evidence to support the notion that PF promotes healthy habits for life in children and adolescents [24]. A research study demonstrated that baseless perspectives or poor knowledge of teachers about fitness development can negatively affect the self-esteem of students and their opinions about PF [25]. Research indicates that fitness tests can be used in negative ways, harming the learning outcomes of students in PE [26].

The PE curriculum can be applied by teachers in different ways, considering either a health-based or an education-based approach. Schools that have a health-based PE curriculum are known for giving students, through PE classes, the chance to be physically active. The school system is important for promoting PA and health among children and adolescents [27]. An education-based PE curriculum can be centred on educational models such as the Comprehensive School Physical Activity Program (CSPAP), which tries to expand the PE program to the school community and help children and adolescents to achieve the recommendations stated above for PA [28].

As the perspectives on fitness development in PE lessons are not consensual among teachers, further research is necessary to comprehend their different opinions. Therefore, in this study we aimed to understand the attitude of Portuguese PE teachers toward PF tests in schools.

## 2. Materials and Methods

### 2.1. Procedures and Participants

This was a cross-sectional study using a large sample of PE teachers. An online survey was conducted between January and March 2019. The survey was disseminated through a mailing contact list of PE teachers. Before completing the survey, informed consent was collected from the teachers. This process resulted in a sample of 764 Portuguese PE teachers (422 male, 342 female), with a mean age of 48.2 years (95% confidence interval (CI) = 47.7, 48.7), teaching at middle- and high-school levels. There were 8306 PE teachers in Portugal at the time of the data collection. Thus, the maximum margin of error associated with a random sample of 764 respondents was 2.8% with a confidence level of 95%. Before collecting the data, the study protocol was approved by the ethics committee of the Faculty of Human Kinetics, University of Lisbon (no. 19/2017), and the Portuguese National Commission for Data Protection (no. 9249/2017).

### 2.2. Measures

The survey included questions about sociodemographic data, teaching, and PA recommendations.

The Physical Education Teacher Attitudes Toward Fitness Tests Scale (PETAFTS) measured the attitudes of the teachers toward fitness [29]. Evidence already exists of the reliability and validity of the PETAFTS in investigating teacher attitudes toward fitness tests. It is a useful instrument for collecting data regarding how teachers perceive the use of fitness tests in schools [30]. The PETAFTS has a 16-item scale consisting of affective and cognitive domains. The affective domain has two subdomains (i.e., enjoyment of implementing fitness tests and using fitness test results). Nine items measure the affective domain and seven measure the cognitive domain. The answer for each item is given on a 7-point Likert scale, with 1 representing “strongly disagree”, 7 representing “strongly agree”, and 4 representing a neutral attitude. The mean score of the affective and cognitive components measures the overall attitude. Thus, a high score indicates a positive attitude.

### 2.3. Statistical Analysis

Descriptive statistics and a 95% CI were calculated for each variable. A one-way ANOVA was used to examine the group differences in the three subdomains of the attitudes of teachers by different variables. A binary logistic regression was performed to analyse the relationship between the overall score of the attitudes of teachers toward fitness and the sociodemographic characteristics using the Enter (Standard) method. The overall score of the attitudes of teachers toward fitness was transformed into a dummy variable. The median value was used to dichotomise the variable and thus analyse the relationship to the value above the median, which indicated a more favourable attitude toward fitness. The fully adjusted model was performed with the attitudes of teachers toward fitness (in its dummy form) as a dependent variable and sex, age group, years of experience, teaching level, and education level of the teachers as covariates. The analysis was performed using IBM SPSS 28. For all analyses, the significance level was set at 5%.

## 3. Results

Table 1 presents the characteristics of the participants. Overall, the attitude of teachers toward fitness tests was slightly positive according to the 7-point Likert scale (5.52, 95% CI: 5.47, 5.58).

The attitude domains of the teachers are presented by characteristics in Table 2. Female teachers had a significantly more positive attitude in the usefulness affective subdomain than male teachers (female: 5.87, 95% CI: 5.78, 5.96; male: 5.67, 95% CI: 5.58, 5.77; *p* = 0.004). However, in the implementation affective subdomain, male teachers demonstrated a significantly more positive attitude toward fitness tests than female teachers (male: 5.51; 95% CI: 5.42, 5.61; female: 5.37, 95% CI: 5.26, 5.47; *p* = 0.042).

Considering the age group variable, younger teachers showed a more positive attitude than older teachers regarding the implementation affective subdomain (younger: 5.52, 95% CI: 5.42, 5.61; older: 5.36, 95% CI: 5.24, 5.47; *p* = 0.029).

Years of experience, teaching level, and education level variables did not reveal statistically significant differences between the groups. No significant interaction effect between the variable groups was found in the cognitive domain and overall score.

Table 3 presents the binary logistic regression results that relate the attitudes of teachers toward fitness tests with their mutually adjusted sociodemographic characteristics. When all the sociodemographic characteristics of the teachers were simultaneously entered into the model, there were no significant relationships. This indicated that there were no differences.

## 4. Discussion

This study examined the attitudes of Portuguese teachers toward PF tests and analysed the differences in the overall attitudes of PE teachers by sex, age group, years of experience, teaching, and education levels. The differences in the attitudes of PE teachers in the cognitive and affective (usefulness and implementation) domains by sex, age group, years of experience, teaching, and education levels were also analysed. PF is an important part of PE and fitness education programs, although it has not always been used to promote health-related fitness [31]. It is also important to mention that researchers have noted that fitness tests can be used positively to enhance the educational experience [32].

### 4.1. Differences in the Overall Attitudes of PE Teachers by Sex, Age Group, Years of Experience, and Teaching and Education Levels

Understanding the beliefs and motives that drive teachers considering the affective or cognitive domains is important. Our results indicated that the attitude of teachers toward fitness tests was slightly positive, agreeing with the existing literature [33]. However, when comparing the overall attitude between groups, there were no significant differences, indicating that Portuguese teachers may show consistency in their general attitude toward fitness tests.

### 4.2. Differences in the Attitude Domains of PE Teachers by Sex, Age Group, Years of Experience, and Teaching and Education Levels

The sex differences in the attitudes of PE teachers toward PF and fitness tests have recently been addressed in the literature. Most studies verified no significant differences in any attitude domain between genders [34]. However, the current study showed interesting findings regarding sex differences in the affective domain. Male teachers enjoyed implementing fitness tests significantly more than female teachers, which corroborated the results of a study in Californian schools [33]. On the other hand, according to their responses, female teachers found fitness tests more useful considering the usefulness of the affective subdomain. Even though we could not determine the reasons for these differences, further investigations should be made to better understand the sex differences in the affective attitude of Portuguese PE teachers. However, when the analysis model was completely adjusted, there were no differences between the male and female teachers. This indicated that sex, combined with other sociodemographic variables, did not seem to distinguish the teachers.

The age group of teachers was a particularly important variable because it could distinguish the attitudes of teachers from different generations with very different educational backgrounds toward fitness tests regarding the importance of PF to children and adolescents. We advise precaution when comparing these findings with existing investigations on this matter because most studies set a lower age limit to determine older and younger teachers. Even though it could be assumed that younger PE teachers had PF concepts more present in their education and, therefore, would have a more positive attitude toward fitness tests, this was not verified in all the attitude domains. The literature also seems to refute this perception [33]. Nonetheless, younger teachers found the implementation of fitness tests significantly more enjoyable than older teachers. However, no differences were observed in the affective usefulness subdomain and the cognitive domain. This could mean that the importance of promoting healthy lifestyles may not be consensually related among teachers to the development of PF through fitness tests.

The results shown in the three other group variables of this study corroborated this last idea. No significantly different attitudes were found in the affective and cognitive domains, distinguishing the teaching level, years of experience, and education level of the teachers. Regarding years of experience, it could be important to investigate why teachers did not develop a more positive attitude toward fitness tests with the increase of years in implementing them.

The lack of significant differences among teachers according to the sociodemographic characteristics in general showed that any differences may be related to the ethos of the schools where they taught or possibly to the training programs they had undertaken [14]. This could mean that, if it is assumed that a physical fitness assessment is important in physical education classes, this concern should begin to be expressed among teachers in training from this early age. In this way, they may more easily develop a favourable attitude toward implementing physical fitness tests in physical education classes.

### 4.3. Study Limitations

This study was transversal and it was impossible to establish causal relationships between the variables. We analysed the attitudes of teachers at a specific time, not considering what happened before and after the application. Therefore, we do not know the reasons for the answers of the teachers and their implications on the group variables. Another issue was the representativeness of the population. As the questionnaires were completed online, the attitude results we collected from the participating teachers were not guaranteed to represent all the existing Portuguese PE teachers. On the other hand, we identified priority groups of PE teachers to implement training actions focused on the importance of developing health-related PF.

## 5. Conclusions

Based on the results from the present study, PE teachers showed a slightly positive attitude toward fitness tests. Female teachers demonstrated a more positive attitude in the affective usefulness subdomain than male teachers. On the other hand, male teachers enjoyed implementing fitness tests significantly more than female teachers. Lastly, younger teachers found the implementation of fitness tests significantly more enjoyable than older teachers. However, there were no significant differences between the teachers according to their sociodemographic characteristics. Future research should prioritise specific intervention content considering gender and the age of teachers to reinforce the development of physical fitness through fitness tests in schools.

## Figures and Tables

**Table 1 children-09-01005-t001:** Participant characteristics.

	% Of Mean (95% CI)
Sex Male Female	55.2 (51.7, 58.8) 44.8 (41.2, 48.3)
Age (years)	48.2 (47.7, 48.7)
Age group ≤ 49 years ≥ 50 years	57.6 (54.1, 61.1) 42.4 (38.9, 45.9)
Years of experience	26.7 (21.2, 32.3)
Years of experience ≤ 19 years 20–29 years ≥ 30 years	26.3 (23.2, 29.4) 49.9 (46.3, 53.4) 23.8 (20.8, 26.8)
Teaching level Middle-school High-school Multiple	49.6 (46.1, 53.2) 12.8 (10.5, 15.2) 37.6 (34.1, 41.0)
Education level Bachelor Master Doctor	69.0 (65.7, 72.3) 29.5 (26.2, 32.7) 1.6 (0.7, 2.5)
Attitudes toward fitness tests Affective domain (usefulness) Affective domain (implementation) Cognitive domain Overall score	5.76 (5.69, 5.83) 5.45 (5.38, 5.52) 5.43 (5.36, 5.50) 5.52 (5.47, 5.58)

CI, confidence interval; PA, physical activity.

**Table 2 children-09-01005-t002:** Physical education attitude domains of teachers by characteristics.

	Affective Domain (Usefulness)	*p*-Value	Affective Domain (Implementation)	*p*-Value	Cognitive Domain	*p*-Value	Overall Score	*p*-Value
Sex Female Male	5.87 (5.78, 5.96) 5.67 (5.58, 5.77)	0.004	5.37 (5.26, 5.47) 5.51 (5.42, 5.61)	0.042	5.48 (5.39, 5.57) 5.39 (5.29, 5.48)	0.158	5.55 (5.47, 5.63) 5.51 (5.42, 5.59)	0.491
Age group ≤ 49 years ≥ 50 years	5.76 (5.67, 5.85) 5.77 (5.67, 5.87)	0.851	5.52 (5.42, 5.61) 5.36 (5.24, 5.47)	0.029	5.40 (5.31, 5.49) 5.47 (5.37, 5.57)	0.345	5.53 (5.46, 5.61) 5.51 (5.42, 5.60)	0.694
Years of experience ≤ 19 years 20–29 years ≥ 30 years	5.71 (5.57, 5.84) 5.73 (5.64, 5.83) 5.88 (5.76, 6.00)	0.141	5.57 (5.43, 5.70) 5.43 (5.33, 5.53) 5.35 (5.21, 5.50)	0.105	5.35 (5.22, 5.48) 5.42 (5.32, 5.52) 5.54 (5.42, 5.66)	0.133	5.52 (5.40, 5.63) 5.51 (5.42, 5.60) 5.57 (5.46, 5.67)	0.716
Teaching level Middle-school High-school Multiple	5.75 (5.65, 5.85) 5.80 (5.62, 5.97) 5.77 (5.67, 5.87)	0.886	5.46 (5.36, 5.57) 5.49 (5.31, 5.67) 5.41 (5.30, 5.53)	0.722	5.42 (5.32, 5.52) 5.52 (5.36, 5.68) 5.41 (5.31, 5.52)	0.595	5.52 (5.43, 5.61) 5.58 (5.44, 5.72) 5.51 (5.42, 5.60)	0.731
Education level Bachelor Master or doctor	5.77 (5.69, 5.85) 5.74 (5.62, 5.86)	0.653	5.48 (5.40, 5.57) 5.37 (5.23, 5.50)	0.144	5.45 (5.37, 5.53) 5.39 (5.28, 5.51)	0.446	5.55 (5.48, 5.62) 5.48 (5.37, 5.59)	0.288

**Table 3 children-09-01005-t003:** Relation between the attitudes of teachers toward physical fitness tests by their characteristics.

	OR (95% CI)	*p*-Value
Sex Female Male	1.00 (ref.) 0.99 (0.75, 1.33)	0.989
Age group ≤ 49 years ≥ 50 years	1.00 (ref.) 1.03 (0.69, 1.54)	0.892
Years of experience ≤ 19 years 20–29 years ≥ 30 years	1.00 (ref.) 0.93 (0.64, 1.34) 0.96 (0.55, 1.68)	0.680 0.881
Teaching level Middle-school High-school Multiple	1.00 (ref.) 0.94 (0.60, 1.49) 0.87 (0.64, 1.19)	0.806 0.373
Education level Bachelor Master or doctor	1.00 (ref.) 1.95 (0.70, 1.30)	0.753

OR, odds ratio; CI, confidence interval.

## Data Availability

The datasets generated and/or analysed during the current study are not publicly available due to the terms of consent/assent to which the participants agreed, but are available from the corresponding author upon reasonable request. Please contact the corresponding author to discuss the availability of the data and materials.

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
