# Peer review of "The Attitude of Portuguese Physical Education Teachers toward Physical Fitness"

_children, 2022, doi:10.3390/children9071005_

Round 1
Reviewer 1 Report
Thank you very much for the opportunity of reviewing this manuscript. The topic and its importance is an issue yet to be discussed among PE/PA teacher/researchers.
However, due the lack of evidence presented and the shortage of the supporting arguments in this paper. I would suggest to modify and improve the way it is presented. It made the impression that the research was straight to the point and the arguments presented made the same. Although it is important to argue more deeply in this regard, since the are opposite points of view whether fitness testing is appropriate for PE classes or not.
For instance in the introduction it is mention that FT should be included in the PE curriculum. However, it is not mention how this should happen. Based on this suggestion, lacks of support whether FT is a viable option to PE. It argues that some PE teacher lack of knowledge of FT. However, it does not discuss whether the PE curriculum should be health-based or education-based, since there are two different approaches and focus that PE teacher can use. Thus, assuming that one is better option than the other one, it a misunderstanding of the issue itself.
Author Response
Comment: Thank you very much for the opportunity of reviewing this manuscript. The topic and its importance is an issue yet to be discussed among PE/PA teacher/researchers.
Response: Thanks for your comment and for your review.
Comment: However, due the lack of evidence presented and the shortage of the supporting arguments in this paper. I would suggest to modify and improve the way it is presented. It made the impression that the research was straight to the point and the arguments presented made the same. Although it is important to argue more deeply in this regard, since the are opposite points of view whether fitness testing is appropriate for PE classes or not.
Response: Thanks for your comment. We look into some aspects of the introduction more deeply in order to add more arguments regarding the use of fitness tests on a global scale and if they are appropriate for PE classes. Even so, for this purpose, we have included more recent references.
Comment: For instance in the introduction it is mention that FT should be included in the PE curriculum. However, it is not mention how this should happen. Based on this suggestion, lacks of support whether FT is a viable option to PE. It argues that some PE teacher lack of knowledge of FT. However, it does not discuss whether the PE curriculum should be health-based or education-based, since there are two different approaches and focus that PE teacher can use. Thus, assuming that one is better option than the other one, it a misunderstanding of the issue itself.
Response: Thank you for bringing up this issue. We addressed the way FT is included in the PE curriculum stating the different fitness tests applied in some countries. Taking into account the two different approaches, we explored the differences between them (regarding the PE curriculum) and stated the main characteristics that diverge them. By doing so, we tried not to favour none so the opinion stays neutral.
Reviewer 2 Report
Thank you for the opportunity to read the article
The theme is very important and adequate after the impact of Pandemic context.
The title is too general. It has to underline that this is a study applied in Portugal.
The Literature review is missing. The authors should study others research on children fitness and teachers attitude to introducing fitness test in schools. They should find different models of implementation in other countries and test /apply it in Portugal. In this regards the references are very few, some of them old and not very representative.
The authors should provide more information regarding the representativeness of the sample. The p values show that the hypothesis can be extrapolated over the entire population only for gender - Affective domain (usefulness and implementation). I haven't seen the Anova table with values between and within groups.... More complex analysis can be done on the data source - correlation, regression, factor analysis, cluster, etc. Thus the result section is incomplete
I would recommend at least 20 new more papers to be read and cited. Please have in mind to be especially WoS, Scopus papers published in the last 3 years.
I hope you will be able to improve the paper because the main idea is very good.
Success!!
Author Response
Comment: Thank you for the opportunity to read the article
Response: Thanks for your comment and for your review.
Comment: The theme is very important and adequate after the impact of Pandemic context.
Response: Thank you for your comment.
Comment: The title is too general. It has to underline that this is a study applied in Portugal.
Response: Thank you. We follow your recommendation. We changed the title to: The attitude of Portuguese physical education teachers’ toward physical fitness. Thus, we got a more specific title.
Comment: The Literature review is missing. The authors should study others research on children fitness and teachers attitude to introducing fitness test in schools. They should find different models of implementation in other countries and test /apply it in Portugal. In this regards the references are very few, some of them old and not very representative.
Response: Thanks for the comment, which set us up to analyse and comprehend the different types of implementations (different test batteries) in other countries. Also, we find very useful and important to explain the application in Portugal to contextualise the aim of the study. We tried to research studies with more representativeness and that were applied in children. As to the teachers' attitude, we did not find more articles with quality than the ones we have, so we focused on the implementation of fitness testing and the controversial ideas regarding it, which was an idea already stated but not deepened.
Comment: The authors should provide more information regarding the representativeness of the sample. The p values show that the hypothesis can be extrapolated over the entire population only for gender - Affective domain (usefulness and implementation). I haven't seen the Anova table with values between and within groups.... More complex analysis can be done on the data source - correlation, regression, factor analysis, cluster, etc. Thus the result section is incomplete
Response: Thanks for the suggestion. In the article we added information about the representativeness of the sample. We performed a binary logistic regression model to analyse the relationship between the overall score of teachers' attitudes toward fitness and sociodemographic characteristics, in a completely adjusted model. The overall score of teachers’ attitudes toward fitness was transformed into a dummy variable. The median value was used to dichotomize the variable and thus analyse the relationship to the value above the median, which means a more favourable attitude toward fitness.
Comment: I would recommend at least 20 new more papers to be read and cited. Please have in mind to be especially WoS, Scopus papers published in the last 3 years.
Response: Thanks for your comment. We explored the suggested databases.
Comment: I hope you will be able to improve the paper because the main idea is very good.
Response: Once again thank you for the comment. Thank you also for the excellence in the review. We hope that we have done what is necessary to improve our article.
Round 2
Reviewer 1 Report
Best of luck in further studies.
Author Response
Thank you for the comment.
Reviewer 2 Report
I would recommend the authors to describe better the regression model.
Most of my concerns were answered.
Congratulation!
Author Response
In the new version of the article, we provide more information about the regression analysis.